# Reversible Recurrent Neural Networks

**Matthew MacKay, Paul Vicol, Jimmy Ba, Roger Grosse**
University of Toronto
Vector Institute
{mmackay, pvicol, jba, rgrosse}@cs.toronto.edu

## Abstract

Recurrent neural networks (RNNs) provide state-of-the-art performance in processing sequential data but are memory intensive to train, limiting the flexibility of RNN models which can be trained. Reversible RNNs—RNNs for which the hidden-to-hidden transition can be reversed—offer a path to reduce the memory requirements of training, as hidden states need not be stored and instead can be recomputed during backpropagation. We first show that perfectly reversible RNNs, which require no storage of the hidden activations, are fundamentally limited because they cannot forget information from their hidden state. We then provide a scheme for storing a small number of bits in order to allow perfect reversal with forgetting. Our method achieves comparable performance to traditional models while reducing the activation memory cost by a factor of 10–15. We extend our technique to attention-based sequence-to-sequence models, where it maintains performance while reducing activation memory cost by a factor of 5–10 in the encoder, and a factor of 10–15 in the decoder.

## 1 Introduction

Recurrent neural networks (RNNs) have attained state-of-the-art performance on a variety of tasks, including speech recognition [Graves et al., 2013], language modeling [Melis et al., 2017, Merity et al., 2017], and machine translation [Bahdanau et al., 2014, Wu et al., 2016]. However, RNNs are memory intensive to train. The standard training algorithm is truncated backpropagation through time (TBPTT) [Werbos, 1990, Rumelhart et al., 1986]. In this algorithm, the input sequence is divided into subsequences of smaller length, say $T$. Each of them is processed and the gradient is backpropagated. If $H$ is the size of our model's hidden state, the memory required for TBPTT is $O(TH)$.

Decreasing the memory requirements of the TBPTT algorithm would allow us to increase the length $T$ of our truncated sequences, capturing dependencies over longer time scales. Alternatively, we could increase the size $H$ of our hidden state or use deeper input-to-hidden, hidden-to-hidden, or hidden-to-output transitions, granting our model greater expressivity. Increasing the depth of these transitions has been shown to increase performance in polyphonic music prediction, language modeling, and neural machine translation (NMT) [Pascanu et al., 2013, Barone et al., 2017, Zilly et al., 2016].

Reversible recurrent network architectures present an enticing way to reduce the memory requirements of TBPTT. Reversible architectures enable the reconstruction of the hidden state at the current timestep given the next hidden state and the current input, which would enable us to perform TBPTT without storing the hidden states at each timestep. In exchange, we pay an increased computational cost to reconstruct the hidden states during backpropagation.

We first present reversible analogues of the widely used Gated Recurrent Unit (GRU) [Cho et al., 2014] and Long Short-Term Memory (LSTM) [Hochreiter and Schmidhuber, 1997] architectures. We then show that any perfectly reversible RNN requiring no storage of hidden activations will fail on a simple one-step prediction task. This task is trivial to solve even for vanilla RNNs, but perfectly reversible models fail since they need to memorize the input sequence in order to solve the task.

In light of this finding, we extend the memory-efficient reversal method of Maclaurin et al. [2015], storing a handful of bits per unit in order to allow perfect reversal for architectures which forget information.

We evaluate the performance of these models on language modeling and neural machine translation benchmarks. Depending on the task, dataset, and chosen architecture, reversible models (without attention) achieve 10–15-fold memory savings over traditional models. Reversible models achieve approximately equivalent performance to traditional LSTM and GRU models on word-level language modeling on the Penn TreeBank dataset [Marcus et al., 1993] and lag 2–5 perplexity points behind traditional models on the WikiText-2 dataset [Merity et al., 2016].

Achieving comparable memory savings with attention-based recurrent sequence-to-sequence models is difficult, since the encoder hidden states must be kept simultaneously in memory in order to perform attention. We address this challenge by performing attention over a small subset of the hidden state, concatenated with the word embedding. With this technique, our reversible models succeed on neural machine translation tasks, outperforming baseline GRU and LSTM models on the Multi30K dataset [Elliott et al., 2016] and achieving competitive performance on the IWSLT 2016 [Cettolo et al., 2016] benchmark. Applying our technique reduces memory cost by a factor of 10–15 in the decoder, and a factor of 5–10 in the encoder.[1]

## 2   Background

We begin by describing techniques to construct reversible neural network architectures, which we then adapt to RNNs. Reversible networks were first motivated by the need for flexible probability distributions with tractable likelihoods [Papamakarios et al., 2017, Dinh et al., 2016, Kingma et al., 2016]. Each of these architectures defines a mapping between probability distributions, one of which has a simple, known density. Because this mapping is reversible with an easily computable Jacobian determinant, maximum likelihood training is efficient.

A recent paper, closely related to our work, showed that reversible network architectures can be adapted to image classification tasks [Gomez et al., 2017]. Their architecture, called the Reversible Residual Network or *RevNet*, is composed of a series of reversible blocks. Each block takes an input $x$ and produces an output $y$ of the same dimensionality. The input $x$ is separated into two groups: $x = [x_1; x_2]$, and outputs are produced according to the following coupling rule:

$$y_1 = x_1 + F(x_2) \qquad y_2 = x_2 + G(y_1) \tag{1}$$

where $F$ and $G$ are residual functions analogous to those in standard residual networks [He et al., 2016]. The output $y$ is formed by concatenating $y_1$ and $y_2$, $y = [y_1; y_2]$. Each layer's activations can be reconstructed from the next layer's activations as follows:

$$x_2 = y_2 - G(y_1) \qquad x_1 = y_1 - F(x_2) \tag{2}$$

Because of this property, activations from the forward pass need not be stored for use in the backwards pass. Instead, starting from the last layer, activations of previous layers are reconstructed during backpropagation[2]. Because reversible backprop requires an additional computation of the residual functions to reconstruct activations, it requires 33% more arithmetic operations than ordinary backprop and is about 50% more expensive in practice. Full details of how to efficiently combine reversibility with backpropagation may be found in Gomez et al. [2017].

## 3   Reversible Recurrent Architectures

The techniques used to construct RevNets can be combined with traditional RNN models to produce reversible RNNs. In this section, we propose reversible analogues of the GRU and the LSTM.

### 3.1   Reversible GRU

We start by recalling the GRU equations used to compute the next hidden state $h^{(t+1)}$ given the current hidden state $h^{(t)}$ and the current input $x^{(t)}$ (omitting biases):

$$[z^{(t)}; r^{(t)}] = \sigma(W[x^{(t)}; h^{(t-1)}]) \qquad g^{(t)} = \tanh(U[x^{(t)}; r^{(t)} \odot h^{(t-1)}])$$
$$h^{(t)} = z^{(t)} \odot h^{(t-1)} + (1 - z^{(t)}) \odot g^{(t)} \tag{3}$$

Here, $\odot$ denotes elementwise multiplication. To make this update reversible, we separate the hidden state $h$ into two groups, $h = [h_1; h_2]$. These groups are updated using the following rules:

$$[z_1^{(t)}; r_1^{(t)}] = \sigma(W_1[x^{(t)}; h_2^{(t-1)}]) \qquad\qquad [z_2^{(t)}; r_2^{(t)}] = \sigma(W_2[x^{(t)}; h_1^{(t)}])$$
$$g_1^{(t)} = \tanh(U_1[x^{(t)}; r_1^{(t)} \odot h_2^{(t-1)}]) \quad (4) \qquad g_2^{(t)} = \tanh(U_2[x^{(t)}; r_2^{(t)} \odot h_1^{(t)}]) \quad (5)$$
$$h_1^{(t)} = z_1^{(t)} \odot h_1^{(t-1)} + (1 - z_1^{(t)}) \odot g_1^{(t)} \qquad h_2^{(t)} = z_2^{(t)} \odot h_2^{(t-1)} + (1 - z_2^{(t)}) \odot g_2^{(t)}$$

Note that $h_1^{(t)}$ and not $h_1^{(t-1)}$ is used to compute the update for $h_2^{(t)}$. We term this model the Reversible Gated Recurrent Unit, or *RevGRU*. Note that $z_i^{(t)} \neq 0$ for $i = 1, 2$ as it is the output of a sigmoid, which maps to the open interval $(0, 1)$. This means the RevGRU updates are reversible in exact arithmetic: given $h^{(t)} = [h_1^{(t)}; h_2^{(t)}]$, we can use $h_1^{(t)}$ and $x^{(t)}$ to find $z_2^{(t)}$, $r_2^{(t)}$, and $g_2^{(t)}$ by redoing part of our forwards computation. Then we can find $h_2^{(t-1)}$ using:

$$h_2^{(t-1)} = [h_2^{(t)} - (1 - z_2^{(t)}) \odot g_2^{(t)}] \odot 1/z_2^{(t)} \tag{6}$$

$h_1^{(t-1)}$ is reconstructed similarly. We address numerical issues which arise in practice in Section 3.3.

## 3.2 Reversible LSTM

We next construct a reversible LSTM. The LSTM separates the hidden state into an output state $h$ and a cell state $c$. The update equations are:

$$[f^{(t)}, i^{(t)}, o^{(t)}] = \sigma(W[x^{(t)}; h^{(t-1)}]) \quad (7) \qquad g^{(t)} = \tanh(U[x^{(t)}; h^{(t-1)}]) \quad (8)$$
$$c^{(t)} = f^{(t)} \odot c^{(t-1)} + i^{(t)} \odot g^{(t)} \quad (9) \qquad\qquad h^{(t)} = o^{(t)} \odot \tanh(c^{(t)}) \quad (10)$$

We cannot straightforwardly apply our reversible techniques, as the update for $h^{(t)}$ is not a non-zero linear transformation of $h^{(t-1)}$. Despite this, reversibility can be achieved using the equations:

$$[f_1^{(t)}, i_1^{(t)}, o_1^{(t)}, p_1^{(t)}] = \sigma(W_1[x^{(t)}; h_2^{(t-1)}]) \quad (11) \qquad g_1^{(t)} = \tanh(U_1[x^{(t)}; h_2^{(t-1)}]) \quad (12)$$
$$c_1^{(t)} = f_1^{(t)} \odot c_1^{(t-1)} + i_1^{(t)} \odot g_1^{(t)} \quad (13) \qquad h_1^{(t)} = p_1^{(t)} \odot h_1^{(t-1)} + o_1^{(t)} \odot \tanh(c_1^{(t)}) \quad (14)$$

We calculate the updates for $c_2, h_2$ in an identical fashion to the above equations, using $c_1^{(t)}$ and $h_1^{(t)}$. We call this model the Reversible LSTM, or *RevLSTM*.

## 3.3 Reversibility in Finite Precision Arithmetic

We have defined RNNs which are reversible in exact arithmetic. In practice, the hidden states cannot be perfectly reconstructed due to finite numerical precision. Consider the RevGRU equations 4 and 5. If the hidden state $h$ is stored in fixed point, multiplication of $h$ by $z$ (whose entries are less than 1) destroys information, preventing perfect reconstruction. Multiplying a hidden unit by $1/2$, for example, corresponds to discarding its least-significant bit, whose value cannot be recovered in the reverse computation. These errors from information loss accumulate exponentially over timesteps, causing the initial hidden state obtained by reversal to be far from the true initial state. The same issue also affects the reconstruction of the RevLSTM hidden states. Hence, we find that forgetting is the main roadblock to constructing perfectly reversible recurrent architectures.

There are two possible avenues to address this limitation. The first is to remove the forgetting step. For the RevGRU, this means we compute $z_i^{(t)}$, $r_i^{(t)}$, and $g_i^{(t)}$ as before, and update $h_i^{(t)}$ using:

$$h_i^{(t)} = h_i^{(t-1)} + (1 - z_i^{(t)}) \odot g_i^{(t)} \tag{15}$$

We term this model the No-Forgetting RevGRU or *NF-RevGRU*. Because the updates of the NF-RevGRU do not discard information, we need only store one hidden state in memory at a given time during training. Similar steps can be taken to define a NF-RevLSTM.

The second avenue is to accept some memory usage and store the information forgotten from the hidden state in the forward pass. We can then achieve perfect reconstruction by restoring this information to our hidden state in the reverse computation. We discuss how to do so efficiently in Section 5.

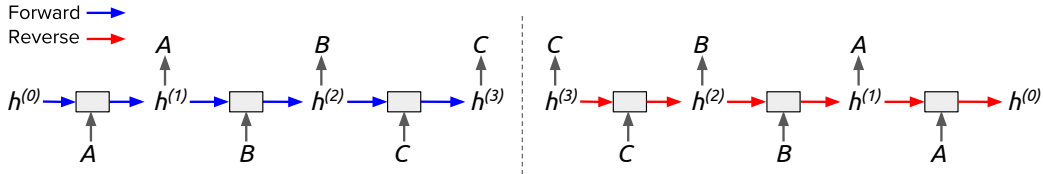

Figure 1: Unrolling the reverse computation of an exactly reversible model on the repeat task yields a sequence-to-sequence computation. **Left:** The repeat task itself, where the model repeats each input token. **Right:** Unrolling the reversal. The model effectively uses the final hidden state to reconstruct all input tokens, implying that the entire input sequence must be stored in the final hidden state.

## 4  Impossibility of No Forgetting

We have shown reversible RNNs in finite precision can be constructed by ensuring that no information is discarded. We were unable to find such an architecture that achieved acceptable performance on tasks such as language modeling[3]. This is consistent with prior work which found forgetting to be crucial to LSTM performance [Gers et al., 1999, Greff et al., 2017]. In this section, we argue that this results from a fundamental limitation of no-forgetting reversible models: if none of the hidden state can be forgotten, then the hidden state at any given timestep must contain enough information to reconstruct all previous hidden states. Thus, any information stored in the hidden state at one timestep must remain present at all future timesteps to ensure exact reconstruction, overwhelming the storage capacity of the model.

We make this intuition concrete by considering an elementary sequence learning task, the *repeat* task. In this task, an RNN is given a sequence of discrete tokens and must simply repeat each token at the subsequent timestep. This task is trivially solvable by ordinary RNN models with only a handful of hidden units, since it doesn't require modeling long-distance dependencies. But consider how an exactly reversible model would perform the repeat task. Unrolling the reverse computation, as shown in Figure 1, reveals a sequence-to-sequence computation in which the encoder and decoder weights are tied. The encoder takes in the tokens and produces a final hidden state. The decoder uses this final hidden state to produce the input sequence in reverse sequential order.

Notice the relationship to another sequence learning task, the *memorization* task, used as part of a curriculum learning strategy by Zaremba and Sutskever [2014]. After an RNN observes an entire sequence of input tokens, it is required to output the input sequence in reverse order. As shown in Figure 1, the memorization task for an ordinary RNN reduces to the repeat task for an NF-RevRNN. Hence, if the memorization task requires a hidden representation size that grows with the sequence length, then so does the repeat task for NF-RevRNNs.

We confirmed experimentally that NF-RevGRU and NF-RevLSM networks with limited capacity were unable to solve the repeat task[4]. Interestingly, the NF-RevGRU was able to memorize input sequences using considerably fewer hidden units than the ordinary GRU or LSTM, suggesting it may be a useful architecture for tasks requiring memorization. Consistent with the results on the repeat task, the NF-RevGRU and NF-RevLSTM were unable to match the performance of even vanilla RNNs on word-level language modeling on the Penn TreeBank dataset [Marcus et al., 1993].

## 5  Reversibility with Forgetting

The impossibility of zero forgetting leads us to explore the second possibility to achieve reversibility: storing information lost from the hidden state during the forward computation, then restoring it in the reverse computation. Initially, we investigated *discrete forgetting*, in which only an integral number of bits are allowed to be forgotten. This leads to a simple implementation: if $n$ bits are forgotten in the forwards pass, we can store these $n$ bits on a stack, to be popped off and restored to the hidden state during reconstruction. However, restricting our model to forget only an integral number of bits led to a substantial drop in performance compared to baseline models[5]. For the remainder of

**Algorithm 1** Exactly reversible multiplication (Maclaurin et al. [2015])
---
1: **Input:** Buffer integer $B$, hidden state $h = 2^{-R_H} h^*$, forget value $z = 2^{-R_Z} z^*$ with $0 < z^* < 2^{R_Z}$
2: $B \leftarrow B \times 2^{R_Z}$ {make room for new information on buffer}
3: $B \leftarrow B + (h^* \mod 2^{R_Z})$ {store lost information in buffer}
4: $h^* \leftarrow h^* \div 2^{R_Z}$ {divide by denominator of $z$}
5: $h^* \leftarrow h^* \times z^*$ {multiply by numerator of $z$}
6: $h^* \leftarrow h^* + (B \mod z^*)$ {add information to hidden state}
7: $B \leftarrow B \div z^*$ {shorten information buffer}
8: **return** updated buffer $B$, updated value $h = 2^{-R_H} h^*$
---

this paper, we turn to *fractional forgetting*, in which a fractional number of bits are allowed to be forgotten.

To allow forgetting of a fractional number of bits, we use a technique introduced by Maclaurin et al. [2015] to store lost information. To avoid cumbersome notation, we do away with super- and subscripts and consider a single hidden unit $h$ and its forget value $z$. We represent $h$ and $z$ as fixed-point numbers (integers with an implied radix point). For clarity, we write $h = 2^{-R_H} h^*$ and $z = 2^{-R_Z} z^*$. Hence, $h^*$ is the number stored on the computer and multiplication by $2^{-R_H}$ supplies the implied radix point. In general, $R_H$ and $R_Z$ are distinct. Our goal is to multiply $h$ by $z$, storing as few bits as necessary to make this operation reversible.

The full process of reversible multiplication is shown in detail in Algorithm 1. The algorithm maintains an integer information buffer which stores $h^* \mod 2^{R_Z}$ at each timestep, so integer division of $h^*$ by $2^{R_Z}$ is reversible. However, this requires enlarging the buffer by $R_Z$ bits at each timestep. Maclaurin et al. [2015] reduced this storage requirement by shifting information from the buffer back onto the hidden state. Reversibility is preserved if the shifted information is small enough so that it does not affect the reverse operation (integer division of $h^*$ by $z^*$). We include a full review of the algorithm of Maclaurin et al. [2015] in Appendix C.1.

However, this trick introduces a new complication not discussed by Maclaurin et al. [2015]: the information shifted from the buffer could introduce significant noise into the hidden state. Shifting information requires adding a positive value less than $z^*$ to $h^*$. Because $z^* \in (0, 2^{R_Z})$ ($z$ is the output of a sigmoid function and $z = 2^{-R_Z} z^*$), $h = 2^{-R_H} h^*$ may be altered by as much $(2^{R_Z} - 1)/2^{R_H}$. If $R_Z \geq R_H$, this can alter the hidden state $h$ by 1 or more[6]. This is substantial, as in practice we observe $|h| \leq 16$. Indeed, we observed severe performance drops for $R_H$ and $R_Z$ close to equal.

The solution is to limit the amount of information moved from the buffer to the hidden state by setting $R_Z$ smaller than $R_H$. We found $R_H = 23$ and $R_Z = 10$ to work well. The amount of noise added onto the hidden state is bounded by $2^{R_Z - R_H}$, so with these values, the hidden state is altered by at most $2^{-13}$. While the precision of our forgetting value $z$ is limited to 10 bits, previous work has found that neural networks can be trained with precision as low as 10–15 bits and reach the same performance as high precision networks [Gupta et al., 2015, Courbariaux et al., 2014]. We find our situation to be similar.

**Memory Savings** To analyze the savings that are theoretically possible using the procedure above, consider an idealized memory buffer which maintains dynamically resizing storage integers $B_h^i$ for each hidden unit $h$ in groups $i = 1, 2$ of the RevGRU model. Using the above procedure, at each timestep the number of bits stored in each $B_h^i$ grows by:

$$R_Z - \log_2(z_{i,h}^*) = \log_2\left(2^{R_Z}/z_{i,h}^*\right) = \log_2\left(1/z_{i,h}\right) \tag{16}$$

If the entries of $z_{i,h}$ are not close to zero, this compares favorably with the naïve cost of 32 bits per timestep. The total storage cost of TBPTT for a RevGRU model with hidden state size $H$ on a sequence of length $T$ will be [7]:

$$-\left[\sum_{t=T}^{T}\sum_{h=1}^{H} \log_2(z_{1,h}^{(t)}) + \log_2(z_{2,h}^{(t)})\right] \tag{17}$$

Thus, in the idealized case, the number of bits stored equals the number of bits forgotten.

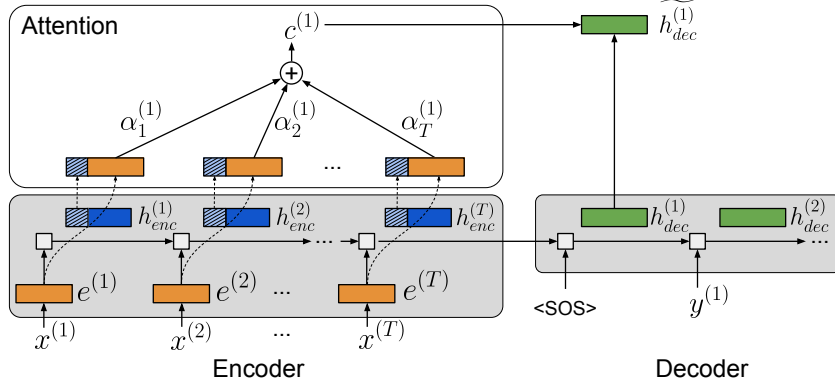

Figure 2: **Attention mechanism for NMT.** The word embeddings, encoder hidden states, and decoder hidden states are color-coded orange, blue, and green, respectively; the striped regions of the encoder hidden states represent the *slices* that are stored in memory for attention. The final vectors used to compute the context vector are concatenations of the word embeddings and encoder hidden state slices.

## 5.1 GPU Considerations

For our method to be used as part of a practical training procedure, we must run it on a parallel architecture such as a GPU. This introduces additional considerations which require modifications to Algorithm 1: (1) we implement it with ordinary finite-bit integers, hence dealing with overflow, and (2) for GPU efficiency, we ensure uniform memory access patterns across all hidden units.

**Overflow.** Consider the storage required for a single hidden unit. Algorithm 1 assumes unboundedly large integers, and hence would need to be implemented using dynamically resizing integer types, as was done by Maclaurin et al. [2015]. But such data structures would require non-uniform memory access patterns, limiting their efficiency on GPU architectures. Therefore, we modify the algorithm to use ordinary finite integers. In particular, instead of a single integer, the buffer is represented with a sequence of 64-bit integers $(B_0, \ldots, B_D)$. Whenever the last integer in our buffer is about to overflow upon multiplication by $2^{R_Z}$, as required by step 1 of Algorithm 1, we append a new integer $B_{D+1}$ to the sequence. Overflow will occur if $B_D > 2^{64-R_Z}$.

After appending a new integer $B_{D+1}$, we apply Algorithm 1 unmodified, using $B_{D+1}$ in place of $B$. It is possible that up to $R_Z - 1$ bits of $B_D$ will not be used, incurring an additional penalty on storage cost. We experimented with several ways of alleviating this penalty but found that none improved significantly over the storage cost of the initial method.

**Vectorization.** Vectorization imposes an additional penalty on storage. For efficient computation, we cannot maintain different size lists as buffers for each hidden unit in a minibatch. Rather, we must store the buffer as a three-dimensional tensor, with dimensions corresponding to the minibatch size, the hidden state size, and the length of the buffer list. This means each list of integers being used as a buffer for a given hidden unit must be the same size. Whenever a buffer being used for any hidden unit in the minibatch overflows, an extra integer must be added to the buffer list for every hidden unit in the minibatch. Otherwise, the steps outlined above can still be followed.

We give the complete, revised algorithm in Appendix C.3. The compromises to address overflow and vectorization entail additional overhead. We measure the size of this overhead in Section 6.

## 5.2 Memory Savings with Attention

Most modern architectures for neural machine translation make use of attention mechanisms [Bahdanau et al., 2014, Wu et al., 2016]; in this section, we describe the modifications that must be made to obtain memory savings when using attention. We denote the source tokens by $x^{(1)}, x^{(2)}, \ldots, x^{(T)}$, and the corresponding word embeddings by $e^{(1)}, e^{(2)}, \ldots, e^{(T)}$. We also use the following notation to denote *vector slices*: given a vector $v \in \mathbb{R}^D$, we let $v[: k] \in \mathbb{R}^k$ denote the vector consisting of the first $k$ dimensions of $v$. Standard attention-based models for NMT perform attention over the encoder hidden states; this is problematic from the standpoint of memory savings, because we must retain the hidden states in memory to use them when computing attention. To remedy this, we explore several alternatives to storing the full hidden state in memory. In particular, we consider performing attention over: 1) the *embeddings* $e^{(t)}$, which capture the semantics of individual words; 2) *slices* of

Table 1: Validation perplexities (memory savings) on Penn TreeBank word-level language modeling. Results shown when forgetting is restricted to 2, 3, and 5 bits per hidden unit per timestep and when there is no restriction.

| Reversible Model | 2 bit | 3 bits | 5 bits | No limit | Usual Model | No limit |
|---|---|---|---|---|---|---|
| 1 layer RevGRU | 82.2 (13.8) | 81.1 (10.8) | 81.1 (7.4) | 81.5 (6.4) | 1 layer GRU | 82.2 |
| 2 layer RevGRU | 83.8 (14.8) | 83.8 (12.0) | 82.2 (9.4) | 82.3 (4.9) | 2 layer GRU | 81.5 |
| 1 layer RevLSTM | 79.8 (13.8) | 79.4 (10.1) | 78.4 (7.4) | 78.2 (4.9) | 1 layer LSTM | 78.0 |
| 2 layer RevLSTM | 74.7 (14.0) | 72.8 (10.0) | 72.9 (7.3) | 72.9 (4.9) | 2 layer LSTM | 73.0 |

the encoder hidden states, $h_{enc}^{(t)}[:k]$ (where we consider $k = 20$ or $100$); and 3) the concatenation of embeddings and hidden state slices, $[e^{(t)}; h_{enc}^{(t)}[:k]]$. Since the embeddings are computed directly from the input tokens, they don't need to be stored. When we slice the hidden state, only the slices that are attended to must be stored. We apply our memory-saving buffer technique to the remaining $D - k$ dimensions.

In our NMT models, we make use of the global attention mechanism introduced by Luong et al. [2015], where each decoder hidden state $h_{dec}^{(t)}$ is modified by incorporating context from the source annotations: a context vector $c^{(t)}$ is computed as a weighted sum of source annotations (with weights $\alpha_j^{(t)}$); $h_{dec}^{(t)}$ and $c^{(t)}$ are used to produce an attentional decoder hidden state $\widetilde{h_{dec}^{(t)}}$. Figure 2 illustrates this attention mechanism, where attention is performed over the concatenated embeddings and hidden state slices. Additional details on attention are provided in Appendix F.

### 5.3 Additional Considerations

**Restricting forgetting.** In order to guarantee memory savings, we may restrict the entries of $z_i^{(t)}$ to lie in $(a, 1)$ rather than $(0, 1)$, for some $a > 0$. Setting $a = 0.5$, for example, forces our model to forget at most one bit from each hidden unit per timestep. This restriction may be accomplished by applying the linear transformation $x \mapsto (1 - a)x + a$ to $z_i^{(t)}$ after its initial computation[8].

**Limitations.** The main flaw of our method is the increased computational cost. We must reconstruct hidden states during the backwards pass and manipulate the buffer at each timestep. We find that each step of reversible backprop takes about 2-3 times as much computation as regular backprop. We believe this overhead could be reduced through careful engineering. We did not observe a slowdown in convergence in terms of number of iterations, so we only pay an increased per-iteration cost.

## 6 Experiments

We evaluated the performance of reversible models on two standard RNN tasks: language modeling and machine translation. We wished to determine how much memory we could save using the techniques we have developed, how these savings compare with those possible using an idealized buffer, and whether these memory savings come at a cost in performance. We also evaluated our proposed attention mechanism on machine translation tasks.

### 6.1 Language Modeling Experiments

We evaluated our one- and two-layer reversible models on word-level language modeling on the Penn Treebank [Marcus et al., 1993] and WikiText-2 [Merity et al., 2016] corpora. In the interest of a fair comparison, we kept architectural and regularization hyperparameters the same between all models and datasets. We regularized the hidden-to-hidden, hidden-to-output, and input-to-hidden connections, as well as the embedding matrix, using various forms of dropout[9]. We used the hyperparameters from Merity et al. [2017]. Details are provided in Appendix G.1. We include training/validation curves for all models in Appendix I.

### 6.1.1 Penn TreeBank Experiments

We conducted experiments on Penn TreeBank to understand the performance of our reversible models, how much restrictions on forgetting affect performance, and what memory savings are achievable.

Table 2: Validation perplexities on WikiText-2 word-level language modeling. Results shown when forgetting is restricted to 2, 3, and 5 bits per hidden unit per timestep and when there is no restriction.

| Reversible Model | 2 bits | 3 bits | 5 bits | No limit | Usual model | No limit |
|---|---|---|---|---|---|---|
| 1 layer RevGRU | 97.7 | 97.2 | 96.3 | 97.1 | 1 layer GRU | 97.8 |
| 2 layer RevGRU | 95.2 | 94.7 | 95.3 | 95.0 | 2 layer GRU | 93.6 |
| 1 layer RevLSTM | 94.8 | 94.5 | 94.5 | 94.1 | 1 layer LSTM | 89.3 |
| 2 layer RevLSTM | 90.7 | 87.7 | 87.0 | 86.0 | 2 layer LSTM | 82.2 |

**Performance.** With no restriction on the amount forgotten, one- and two-layer RevGRU and RevLSTM models obtained roughly equivalent validation performance[10] compared to their non-reversible counterparts, as shown in Table 1. To determine how little could be forgotten without affecting performance, we also experimented with restricting forgetting to at most 2, 3, or 5 bits per hidden unit per timestep using the method of Section 5.3. Restricting the amount of forgetting to 2, 3, or 5 bits from each hidden unit did not significantly impact performance. Performance suffered once forgetting was restricted to at most 1 bit. This caused a 4–5 increase in perplexity for the RevGRU. It also made the RevLSTM unstable for this task since its hidden state, unlike the RevGRU's, can grow unboundedly if not enough is forgotten. Hence, we do not include these results.

**Memory savings.** We tracked the size of the information buffer throughout training and used this to compare the memory required when using reversibility vs. storing all activations. As shown in Appendix H, the buffer size remains roughly constant throughout training. Therefore, we show the average ratio of memory requirements during training in Table 1. Overall, we can achieve a 10–15-fold reduction in memory when forgetting at most 2–3 bits, while maintaining comparable performance to standard models. Using Equation 17, we also compared the actual memory savings to the idealized memory savings possible with a perfect buffer. In general, we use about twice the amount of memory as theoretically possible. Plots of memory savings for all models, both idealized and actual, are given in Appendix H.

### 6.1.2 WikiText-2 Experiments

We conducted experiments on the WikiText-2 dataset (WT2) to see how reversible models fare on a larger, more challenging dataset. We investigated various restrictions, as well as no restriction, on forgetting and contrasted with baseline models as shown in Table 2. The RevGRU model matched the performance of the baseline GRU model, even with forgetting restricted to 2 bits. The RevLSTM lagged behind the baseline LSTM by about 5 perplexity points for one- and two-layer models.

### 6.2 Neural Machine Translation Experiments

We further evaluated our models on English-to-German neural machine translation (NMT). We used a unidirectional encoder-decoder model and our novel attention mechanism described in Section 5.2. We experimented on two datasets: Multi30K [Elliott et al., 2016], a dataset of ~30,000 sentence pairs derived from Flickr image captions, and IWSLT 2016 [Cettolo et al., 2016], a larger dataset of ~180,000 pairs. Experimental details are provided in Appendix G.2; training and validation curves are shown in Appendix I.3 (Multi30K) and I.4 (IWSLT); plots of memory savings during training are shown in Appendix H.2.

For Multi30K, we used single-layer RNNs with 300-dimensional hidden states and 300-dimensional word embeddings for both the encoder and decoder. Our baseline GRU and LSTM models achieved test BLEU scores of 32.60 and 37.06, respectively. The test BLEU scores and encoder memory savings achieved by our reversible models are shown in Table 3, for several variants of attention and restrictions on forgetting. For attention, we use Emb to denote word embeddings, $x$H for a $x$-dimensional slice of the hidden state (300H denotes the whole hidden state), and Emb+$x$H to denote the concatenation of the two. Overall, while Emb attention achieved the best memory savings, Emb+20H achieved the best balance between performance and memory savings. The RevGRU with Emb+20H attention and forgetting at most 2 bits achieved a test BLEU score of 34.41, outperforming the standard GRU, while reducing activation memory requirements by $7.1\times$ and $14.8\times$ in the encoder and decoder, respectively. The RevLSTM with Emb+20H attention and forgetting at most 3 bits achieved a test BLEU score of 37.23, outperforming the standard LSTM, while reducing activation memory requirements by $8.9\times$ and $11.1\times$ in the encoder and decoder respectively.

Table 3: Performance on the Multi30K dataset with different restrictions on forgetting. **P** denotes the test BLEU scores; **M** denotes the average memory savings of the encoder during training.

| Model | Attention | 1 bit | | 2 bit | | 3 bit | | 5 bit | | No Limit | |
|-------|-----------|-------|-------|-------|-------|-------|-------|-------|-------|----------|-------|
| | | **P** | **M** | **P** | **M** | **P** | **M** | **P** | **M** | **P** | **M** |
| RevLSTM | 20H | 29.18 | 11.8 | 30.63 | 9.5 | 30.47 | 8.5 | 30.02 | 7.3 | 29.13 | 6.1 |
| | 100H | 27.90 | 4.9 | 35.43 | 4.3 | 36.03 | 4.0 | 35.75 | 3.7 | 34.96 | 3.5 |
| | 300H | 26.44 | 1.0 | 36.10 | 1.0 | 37.05 | 1.0 | 37.30 | 1.0 | 36.80 | 1.0 |
| | Emb | 31.92 | 20.0 | 31.98 | 15.1 | 31.60 | 13.9 | 31.42 | 10.7 | 31.45 | 10.1 |
| | Emb+20H | 36.80 | 12.1 | 36.78 | 9.9 | 37.23 | 8.9 | 36.45 | 8.1 | 37.30 | 7.4 |
| RevGRU | 20H | 26.52 | 7.2 | 26.86 | 7.2 | 28.26 | 6.8 | 27.71 | 6.5 | 27.86 | 5.7 |
| | 100H | 33.28 | 2.6 | 32.53 | 2.6 | 31.44 | 2.5 | 31.60 | 2.4 | 31.66 | 2.3 |
| | 300H | 34.86 | 1.0 | 33.49 | 1.0 | 33.01 | 1.0 | 33.03 | 1.0 | 33.08 | 1.0 |
| | Emb | 28.51 | 13.2 | 28.76 | 13.2 | 28.86 | 12.9 | 27.93 | 12.8 | 28.59 | 12.9 |
| | Emb+20H | 34.00 | 7.2 | 34.41 | 7.1 | 34.39 | 6.4 | 34.04 | 5.9 | 34.94 | 5.7 |

For IWSLT 2016, we used 2-layer RNNs with 600-dimensional hidden states and 600-dimensional word embeddings for the encoder and decoder. We evaluated reversible models in which the decoder used Emb+60H attention. The baseline GRU and LSTM models achieved test BLEU scores of 16.07 and 22.35, respectively. The RevGRU achieved a test BLEU score of 20.70, outperforming the GRU, while saving $7.15\times$ and $12.92\times$ in the encoder and decoder, respectively. The RevLSTM achieved a score of 22.34, competitive with the LSTM, while saving $8.32\times$ and $6.57\times$ memory in the encoder and decoder, respectively. Both reversible models were restricted to forget at most 5 bits.

## 7   Related Work

Several approaches have been taken to reduce the memory requirements of RNNs. Frameworks that use static computational graphs [Abadi et al., 2016, Al-Rfou et al., 2016] aim to allocate memory efficiently in the training algorithms themselves. Checkpointing [Martens and Sutskever, 2012, Chen et al., 2016, Gruslys et al., 2016] is a frequently used method. In this strategy, certain activations are stored as checkpoints throughout training and the remaining activations are recomputed as needed in the backwards pass. Checkpointing has previously been used to train recurrent neural networks on sequences of length $T$ by storing the activations every $\lceil\sqrt{T}\rceil$ layers [Martens and Sutskever, 2012]. Gruslys et al. [2016] further developed this strategy by using dynamic programming to determine which activations to store in order to minimize computation for a given storage budget.

Decoupled neural interfaces [Jaderberg et al., 2017, Czarnecki et al., 2017] use auxilliary neural networks trained to produce the gradient of a layer's weight matrix given the layer's activations as input, then use these predictions to train, rather than the true gradient. This strategy depends on the quality of the gradient approximation produced by the auxilliary network. Hidden activations must still be stored as in the usual backpropagation algorithm to train the auxilliary networks, unlike our method.

Unitary recurrent neural networks [Arjovsky et al., 2016, Wisdom et al., 2016, Jing et al., 2016] refine vanilla RNNs by parametrizing their transition matrix to be unitary. These networks are reversible in exact arithmetic [Arjovsky et al., 2016]: the conjugate transpose of the transition matrix is its inverse, so the hidden-to-hidden transition is reversible. In practice, this method would run into numerical precision issues as floating point errors accumulate over timesteps. Our method, through storage of lost information, avoids these issues.

## 8   Conclusion

We have introduced reversible recurrent neural networks as a method to reduce the memory requirements of truncated backpropagation through time. We demonstrated the flaws of exactly reversible RNNs, and developed methods to efficiently store information lost during the hidden-to-hidden transition, allowing us to reverse the transition during backpropagation. Reversible models can achieve roughly equivalent performance to standard models while reducing the memory requirements by a factor of 5–15 during training. We believe reversible models offer a compelling path towards constructing more flexible and expressive recurrent neural networks.

**Acknowledgments**

We thank Kyunghyun Cho for experimental advice and discussion. We also thank Aidan Gomez, Mengye Ren, Gennady Pekhimenko, and David Duvenaud for helpful discussion. MM is supported by an NSERC CGS-M award, and PV is supported by an NSERC PGS-D award.

## Footnotes

[1]Code will be made available at `https://github.com/matthewjmackay/reversible-rnn`

[2]The activations prior to a pooling step must still be saved, since this involves projection to a lower dimensional space, and hence loss of information.

[3]We discuss our failed attempts in Appendix A.

[4]We include full results and details in Appendix B. The argument presented applies to idealized RNNs able to implement any hidden-to-hidden transition and whose hidden units can store 32 bits each. We chose to use the LSTM and the NF-RevGRU as approximations to these idealized models since they performed best at their respective tasks.

[5]Algorithmic details and experimental results for discrete forgetting are given in Appendix D.

[6]We illustrate this phenomenon with a concrete example in Appendix C.2.

[7]For the RevLSTM, we would sum over $p_i^{(t)}$ and $f_i^{(t)}$ terms.

[8]For the RevLSTM, we would apply this transformation to $p_i^{(t)}$ and $f_i^{(t)}$.

[9]We discuss why dropout does not require additional storage in Appendix E.

[10]Test perplexities exhibit similar patterns but are 3–5 perplexity points lower.

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
