[Supplementary Material]

## Appendix

Here, we provide additional details about our models and results. This appendix is structured as follows:

- We discuss no-forgetting failures in Sec. A.
- We present results for our toy memorization experiment in Sec. B.
- We provide details on reversible multiplication in Sec. C.
- We discuss discrete forgetting in Sec. D.
- We discuss reversibility with dropout in Sec. E.
- We provide details about the attention mechanism we use in Sec. F.
- We provide details on our language modeling (LM) and neural machine translation (NMT) experiments in Sec. G.
- We plot the memory savings during training for many configurations of our RevGRU and RevLSTM models on LM and NMT in Sec. H.
- We provide training and validation curves for each model on the Penn TreeBank and WikiText2 language modeling task, and on the Multi30K and IWSLT-2016 NMT tasks in Sec. I.

## A    No-Forgetting Failures

We tried training NF-RevGRU models on the Penn TreeBank dataset. Without regularization, the training loss (not perplexity) of NF models blows up and remains above 100. This is because the norm of the hidden state grows very quickly. We tried many techniques to remedy this, including: 1) penalizing the hidden state norm; 2) using different optimizers; 3) using layer normalization; and 4) using better initialization. The best-performing model we found reached 110 train perplexity on PTB without any regularization; in contrast, even heavily regularized baseline models can reach 50 train perplexity.

## B    Toy Task Experiment

We trained an LSTM on the memorization task and an NF-RevGRU on the repeat task on sequences of length $20, 35,$ and $50$. To vary the complexity of the tasks, we experimented with hidden state sizes of $8, 16$ and $32$. We trained on randomly generated synthetic sequences consisting of 8 possible input tokens. To evaluate performance, we generated an evaluation batch of $10, 000$ randomly generated sequences and report the average number of tokens correctly predicted over all sequences in this batch. To ensure exact reversibility of the NF-RevGRU, we used a fixed point representation of the hidden state, while activations were computed in floating point.

Each token was input to the model as a one-hot vector. For the remember task, we appended another category to these one-hot vectors indicating whether the end of the input sequence has occurred. This category was set to $0$ before the input sequence terminated and was $1$ afterwards. Models were trained by a standard cross-entropy loss objective.

We used the Adam optimizer [Kingma and Ba, 2014] with learning rate $0.001$. We found that a large batch size of $20, 000$ was needed to achieve the best performance. We noticed that performance continued to improve, albeit slowly, over long periods of time, so we trained our models for 1 million batches. We report the maximum number of tokens predicted correctly over the course of training, as there are slight fluctuations in evaluation performance during training.

We found a surprisingly large difference in performance between the two tasks, as shown in Table 1. In particular, the NF-RevGRU was able to correctly predict more tokens than expected, indicating that it was able to store a surprising amount of information in its hidden state. We suspect that the NF-RevGRU learns how to compress information more easily than an LSTM. The function NF-RevGRU must learn for the repeat task is inherently local, in contrast to the function the LSTM must learn for the remember task, which has long term dependencies.

---
**Algorithm 1** Exactly reversible multiplication [Maclaurin et al., 2015]
---
1: **Input:** Buffer integer $B$, hidden state $h = 2^{-R_H} h^*$, forget value $z = 2^{-R_z} z^*$ with $0 < z^* < 2^{R_z}$
2: $B \leftarrow B \times 2^{R_z}$ {make room for new information on buffer}
3: $B \leftarrow B + (h^* \bmod 2^{R_z})$ {store lost information in buffer}
4: $h^* \leftarrow h^* \div 2^{R_z}$ {divide by denominator of $z$}
5: $h^* \leftarrow h^* \times z^*$ {multiply by numerator of $z$}
6: $h^* \leftarrow h^* + (B \bmod z^*)$ {add information to hidden state}
7: $B \leftarrow B \div z^*$ {shorten information buffer}
8: **return** updated buffer $B$, updated value $h = 2^{-R_H} h^*$
---

Table 1: Number of correct predictions made by an exactly reversible model, which cannot forget, on the repeat task and a traditional model, which can forget, on the memorization task. We expect these models to achieve equivalent performance given the same hidden state size and sequence length. With random guessing, both models would be expected to correctly predict Sequence Length/8 tokens. We also include the number of bits stored per hidden unit, after subtracting out chance accuracy.

| Hidden Units | Sequence Length | Repeat (NF-RevGRU) | | Memorization (LSTM) | |
|---|---|---|---|---|---|
| | | Tokens predicted | Bits/units | Tokens predicted | Bits/unit |
| 8 | 20 | 7.9 | 2.0 | 7.4 | 1.8 |
| | 35 | 13.1 | 3.3 | 9.7 | 2.0 |
| | 50 | 18.6 | 4.6 | 13.0 | 2.5 |
| 16 | 20 | 19.9 | 3.3 | 13.7 | 2.1 |
| | 35 | 25.4 | 3.9 | 14.3 | 1.9 |
| | 50 | 27.3 | 3.9 | 17.2 | 2.1 |
| 32 | 20 | 20.0 | 2.6 | 20.0 | 2.6 |
| | 35 | 35.0 | 5.6 | 20.6 | 2.4 |
| | 50 | 47.9 | 6.2 | 21.5 | 2.3 |

## C  Reversible Multiplication

### C.1  Review of Algorithm of Maclaurin et al. [2015]

We restate the algorithm of Maclaurin et al. [2015] above for convenience. Recall the goal is to multiply $h = 2^{-R_H} h^*$ by $z = 2^{-R_z} z^*$, storing as few bits as necessary to make this operation reversible. This multiplication is accomplished by first dividing $h^*$ by $2^{R_z}$ then multiplying by $z^*$.

First, observe that integer division of $h^*$ by $2^{R_z}$ can be made reversible through knowledge of $h^* \bmod 2^{R_z}$:

$$h^* = (h^* \div 2^{R_z}) \times 2^{R_z} + (h^* \bmod 2^{R_z}) \tag{1}$$

Thus, the remainders at each timestep must be stored in order to ensure reversibility. The remainders could be stored as separate integers, but this would entail 32 bits of storage at each timestep. Instead, the remainders are stored in a single integer information buffer $B$, which is assumed to dynamically resize upon overflow. At each timestep, the buffer's size must be enlarged by $R_Z$ bits to make room:

$$B \leftarrow B \times 2^{R_z} \tag{2}$$

Then a new remainder can be added to the buffer:

$$B \leftarrow B + (h^* \bmod 2^{R_z}) \tag{3}$$

The storage cost has been reduced from 32 bits to $R_Z$ bits per timestep, but even further savings can be realized. Upon multiplying $h^*$ by $z^*$, there is an opportunity to add an integer $e \in \{0, 1, \ldots, z^* - 1\}$ to $h^*$ without affecting the reverse process (integer division by $z^*$):

$$h^* = (h^* \times z^* + e) \div z^* \tag{4}$$

Maclaurin et al. [2015] took advantage of this and moved information from the buffer $B$ to $h^*$ by adding $B \bmod z^*$ to $h^*$. This allows division $B$ by $z^*$ since this division can be reversed by

---
**Algorithm 2** Reverse process of Maclaurin et al. [2015] Algorithm
---
1: **Input:** Updated buffer integer $B$, updated hidden state $h = 2^{-R_H} h^*$, forget value $z = 2^{-R_Z} z^*$ with
   $0 < z^* < 2^{R_Z}$
2: $B \leftarrow B \times z^*$
3: $B \leftarrow B + (h^* \bmod z^*)$
4: $h^* \leftarrow h^* \div z^*$
5: $h^* \leftarrow h^* \times 2^{R_Z}$
6: $h^* \leftarrow h^* + (B \bmod 2^{R_Z})$
7: $B \leftarrow B \div 2^{R_Z}$
8: **return** Original buffer $B$, original hidden state $h = 2^{-R_H} h^*$
---

knowledge of the modulus $B \bmod z^*$, which can be recovered from $h^*$ in the reverse process:

$$h^* \leftarrow h^* + (B \bmod z^*) \qquad (5)$$
$$B \leftarrow B \div z^* \qquad (6)$$

We give the complete reversal algorithm as Algorithm 2.

### C.2  Noise in Buffer Computations

Suppose we have $R_H = R_Z = 4$, $h^* = 16$, $z^* = 17$ and $B = 1$. We hope to compute the new value
for $h$ of $h = \frac{h^*}{2^{R_H}} \times \frac{z^*}{2^{R_Z}} = \frac{17}{16} = 1.0625$. Executing Algorithm 1 we have:

$$B \leftarrow B \times 2^{R_Z} = 16$$
$$B \leftarrow B + (h^* \bmod 2^{R_Z}) = 16$$
$$h^* \leftarrow h^* \div 2^{R_Z} = 1$$
$$h^* \leftarrow h^* \times z^* = 17$$
$$h^* \leftarrow h^* + (B \bmod 17) = 33$$
$$B \leftarrow B \div z^* = 0$$

At the conclusion of the algorithm, we have that $h = \frac{h^*}{2^{R_Z}} = \frac{33}{16} = 2.0625$. The addition of
information from the buffer onto the hidden state has altered it from its intended value.

### C.3  Vectorized reversible multiplication

We let $N$ denote the current minibatch size. Algorithm 3 shows the vectorized reversible multiplica-
tion.

---
**Algorithm 3** Exactly reversible multiplication with overflow
---
1: **Input:** Hidden state $h = 2^{-R_H} h^*$ with dimensions $(N, H)$; forget value $z = 2^{-R_Z} z^*$ with $0 < z^* < 2^{R_Z}$
   and dimensions $(N, H)$; current buffer $B$, an integer tensor with dimensions $(N, H)$; past buffers $B_{past}$,
   an integer tensor with dimensions $(N, H, D)$
2: **if** any entry of $B$ is $\geq 2^{64-R_Z}$ **then**
3:   $B_{past} \leftarrow [B_{past}, B]$ {Append $B$ to end of $B_{past}$}
4:   $B \leftarrow$ tensor of zeroes with dimensions $(N, H)$ {Initialize new buffer}
5: **end if**
6: Execute Algorithm 1 unchanged
7: **return** updated buffer $B$, updated past buffers $B_{past}$, updated value $h$
---

## D  Discrete Forgetting

### D.1  Description

Here, we consider forgetting a discrete number of bits at each timestep. This is much easier to
implement than fractional forgetting, and it is interesting to explore whether fractional forgetting is
necessary or if discrete forgetting will suffice.

| Model | One layer | | | | | Two layers | | | | |
|---|---|---|---|---|---|---|---|---|---|---|
| | 1 bit | 2 bits | 3 bits | 5 bits | No limit | 1 bit | 2 bits | 3 bits | 5 bits | No limit |
| GRU | - | - | - | - | 82.2 | - | - | - | - | 81.5 |
| DF-RevGRU | 93.6 | 94.1 | 93.9 | 94.7 | - | 93.5 | 92.0 | 93.1 | 94.3 | - |
| FF-RevGRU | 86.0 | 82.2 | 81.1 | 81.1 | 81.5 | 87.0 | 83.8 | 83.8 | 82.2 | 82.3 |
| LSTM | - | - | - | - | 78.0 | - | - | - | - | 73.0 |
| DF-RevLSTM | 85.4 | 85.1 | 86.1 | 86.8 | - | 78.1 | 78.3 | 79.1 | 78.6 | - |
| FF-RevLSTM | - | 79.8 | 79.4 | 78.4 | 78.2 | - | 74.7 | 72.8 | 72.9 | 72.9 |

Table 2: Validation perplexities on Penn TreeBank word-level language modeling. Test perplexities exhibit a similar pattern but are 3–5 perplexity points lower. DF denotes discrete forgetting and FF denotes fractional forgetting. We show perplexities when forgetting is restricted to 1, 2, 3, and 5 bits per hidden unit and when there is no limit placed on the amount forgotten.

Recall that the RevGRU updates proposed in Equations 4 and 5. If all entries of $z_i$ are non-positive powers of 2, then multiplication by $z_i$ corresponds exactly to a right-shift of the bits of $h_i$[1]. The shifted off bits can be stored in a stack, to be popped off and restored in the reverse computation. We enforce this condition by changing the equation computing $z_i$. We first choose the largest negative power of 2 that $z_i$ could possibly represent, say $F$. $z_1^{(t)}$ is computed using[2]:

$$
s_1^{(t)}[i, j] = \text{ReLU}(Q[x^{(t)}, h_2^{(t-1)}])[Hi + j] \text{ for } 1 \leq i \leq H, 1 \leq j \leq F
$$
$$
o_1^{(t)} = \text{Softmax}(\text{SampleOneHot}(s_1^{(t)})) \qquad z_1^{(t)} = [1, 0.5, 0.25, \dots, 2^{-F}] \cdot o_1^{(t)} \tag{7}
$$

The equations to calculate $z_2^{(t)}$ are analogous. We use similar equations to compute $f_i^{(t)}, p_i^{(t)}$ for the RevLSTM. To train these models, we must use techniques to estimate gradients of functions of discrete random variables. We used both the Straight-Through Categorical estimator [Bengio et al., 2013] and the Straight-Through Gumbel-Softmax estimator [Jang et al., 2016, Maddison et al., 2016]. In both these estimators, the forward pass is discretized but gradients during backpropagation are computed as if a continuous sample were used.

The memory savings this represents over traditional models depends on the maximum number of bits $F$ allowed to be forgotten. Instead of storing 32 bits for hidden unit per timestep, we must instead only store at most $F$ bits. We do so by using a list of integers $B = (B_1, B_2, \dots, B_D)$ as an information buffer. To store $n$ bits in $B$, we shift the bits of each $B_i$ left by $n$, then add the $n$ bits to be stored onto $B_1$. We move the bits shifted off of $B_i$ onto $B_{i+1}$ for $i \in \{1, \dots, D - 1\}$. If stored bits are shifted off of $B_D$, we must append another integer to $B$. In practice, we store $F$ bits for each hidden unit regardless of its corresponding forget value. This stores some extraneous bits but is much easier to implement when vectorizing over the hidden unit dimension and the batch dimension on the GPU, as is required for computational efficiency.

## D.2 Experiments

For discrete forgetting, we found the Straight-Through Gumbel-Softmax gradient estimator to consistently achieve results 2–3 perplexity better than the Straight-Through categorical estimator. Hence, all discrete forgetting models whose results are reported were trained using the Straight-Through Gumbel-Softmax estimator.

**Discrete vs. Fractional Forgetting.** We show complete results on Penn TreeBank validation perplexity in Table 2. Overall, models which use discrete forgetting performed 4-10 perplexity points worse on the validation set than their fractional forgetting counterparts. It could be the case that the stochasticity of the samples used in discrete forgetting models already imposes a regularizing effect, causing discrete models to be too heavily regularized. To check, we also ran experiments using lower dropout rates and found that discrete forgetting models still lagged behind their fractional counterparts. We conclude that information must be discarded from the hidden state in fine, not coarse, quantities.

# E  Discussion of Dropout

First, consider dropping out elements of the input. If the same elements are dropped out at each step, we simply store the single mask used, then apply it to the input at each step of our forwards and reverse computation.

Applying dropout to the hidden state does not entail information loss (and hence additional storage), since we can interpret dropout as masking out elements of the input/hidden-to-hidden matrices. If the same dropout masks are used at each timestep, as is commonly done in RNNs, we store the single weight mask used, then use the dropped-out matrix in the forward and reverse passes. If the same rows of these matrices are dropped out (as in variational dropout), we need only store a mask the same size as the hidden state.

If we wish to sample different dropout masks at each timestep, which is not commonly done in RNNs, we would either need to store the mask used at each timestep, which is memory intensive, or devise a way to recover the sampled mask in the reverse computation (e.g., using a reversible sampler, or using a deterministic function to set the random seed at each step).

# F  Attention Details

In our NMT experiments, we use the *global attention mechanism* introduced by Luong et al. [2015]. We consider attention performed over a set of *source-side annotations* $\{s^{(1)}, \ldots, s^{(T)}\}$, which can be either: 1) the encoder hidden states, $s^{(t)} = h_{enc}^{(t)}$; 2) the source embeddings, $s^{(t)} = e^{(t)}$; or 3) a concatenation of the embeddings and $k$-dimensional slices of the hidden states, $s^{(t)} = [e^{(t)}; h_{enc}^{(t)}[: k]]$. When using global attention, the model first computes the decoder hidden states $\{h_{dec}^{(1)}, \ldots, h_{dec}^{(M)}\}$ as in the standard encoder-decoder paradigm, and then it *modifies* each $h_{dec}^{(t)}$ by incorporating context from the source annotations. A context vector $c^{(t)}$ is computed as a weighted sum of the source annotations:

$$c^{(t)} = \sum_{j=1}^{T} \alpha_j^{(t)} s^{(j)} \tag{8}$$

where the weights $\alpha_j^{(t)}$ are computed by scoring the similarity between the "current" decoder hidden state $h_{dec}^{(t)}$ and each of the encoder annotations:

$$\alpha_j^{(t)} = \frac{\exp(\text{score}(h_{dec}^{(t)}, s^{(j)}))}{\sum_{k=1}^{T} \exp(\text{score}(h_{dec}^{(t)}, s^{(k)}))} \tag{9}$$

As the score function, we use the "general" formulation proposed by Luong et al. [2015]:

$$\text{score}(h_{dec}^{(t)}, s^{(j)}) = (h_{dec}^{(t)})^{\top} W_a s^{(j)} \tag{10}$$

Then, the original decoder hidden state $h_{dec}^{(t)}$ is modified via the context $c^{(t)}$, to produce an *attentional* hidden state $\widetilde{h_{dec}^{(t)}}$:

$$\widetilde{h_{dec}^{(t)}} = \tanh(W_c[c^{(t)}; h_{dec}^{(t)}]) \tag{11}$$

Finally, the attentional hidden state $\widetilde{h_{dec}^{(t)}}$ is passed into the softmax layer to produce the output distribution:

$$p(y^{(t)} \mid y^{(1)}, \ldots, y^{(t-1)}, \mathbf{x}) = \text{softmax}\left(W_s \widetilde{h_{dec}^{(t)}}\right) \tag{12}$$

# G  Experiment Details

All experiments were implemented using PyTorch [Paszke et al., 2017]. Neural machine translation experiments were implemented using OpenNMT [Klein et al., 2017].

Table 3: Total number of parameters in each model used for LM.

| Model | Total number of parameters |
|---|---|
| 1 layer GRU | 9.0M |
| 1 layer RevGRU | 8.4M |
| 1 layer LSTM | 9.9M |
| 1 layer RevLSTM | 9.7M |
| 2 layer GRU | 16.2M |
| 2 layer RevGRU | 13.6M |
| 2 layer LSTM | 19.5M |
| 2 layer RevLSTM | 18.4M |

## G.1 Language Modeling Experiments

We largely followed Merity et al. [2017] in setting hyperparameters. All one-layer models used $650$ hidden units and all two-layer models used $1150$ hidden units in their first layer and $650$ in their second. We kept our embedding size constant at $650$ through all experiments.

Notice that with a fixed hidden state size, a reversible architecture will have fewer parameters than a standard architecture. If the total number of hidden units is $H$, the number of hidden-to-hidden parameters is $2 \times (H/2)^2 = H^2/2$ in a reversible model, compared to $H^2$ for its non-reversible counterpart. For the RevLSTM, there are extra hidden-to-hidden parameters due to the $p$ gate needed for reversibility. Each model also has additional parameters associated with the input-to-hidden connections and embedding matrix.

We show the total number of parameters in each model, including embeddings, in Table 3.

We used DropConnect [Wan et al., 2013] with probability $0.5$ to regularize all hidden-to-hidden matrices. We applied variational dropout [Gal and Ghahramani, 2016] on the inputs and outputs of the RNNs. The inputs to the first layer were dropped out with probability $0.3$. The outputs of each layer were dropped out with probability $0.4$. As in Gal and Ghahramani [2016], we used embedding dropout with probability $0.1$. We also applied weight decay with scalar factor $1.2 \times 10^{-6}$.

We used a learning rate of $20$ for all models, clipping the norm of the gradients to be smaller than $0.1$. We decayed the learning rate by a factor of $4$ once the nonmonotonic criterion introduced by Merity et al. [2017] was triggered and used the same non-monotone interval of $5$ epochs. For discrete forgetting models, we found that a learning rate decay factor of $2$ worked better. Training was stopped once the learning rate is below $10^{-2}$.

Like Merity et al. [2017], we used variable length backpropagation sequences. The base sequence length was set to $70$ with probability $0.95$ and set to $35$ otherwise. The actual sequence length used was then computed by adding random noise from $\mathcal{N}(0, 5)$ to the base sequence length. We rescaled the learning rate linearly based on the length of the truncated sequences, so for a given minibatch of length $T$, the learning rate used was $20 \times \frac{T}{70}$.

## G.2 Neural Machine Translation Experiments

**Multi30K Experiments.** The Multi30K dataset [Elliott et al., 2016] contains English-German sentence pairs derived from captions of Flickr images, and consists of 29,000 training, 1,015 validation, and 1,000 test sentence pairs. The average length of the source (English) sequences is 13 tokens, and the average length of the target (German) sequences is 12.4 tokens.

We applied variational dropout with probability $0.4$ to inputs and outputs. We trained on minibatches of size 64 using SGD. The learning rate was initialized to $0.2$ for GRU and RevGRU, $0.5$ for RevLSTM, and $1$ for the standard LSTM—these values were chosen to optimize the performance of each model. The learning rate was decayed by a factor of 2 each epoch when the validation loss failed to improve from the previous epoch. Training halted when the learning rate dropped below $0.001$. Table 4 shows the validation BLEU scores of each RevGRU and RevLSTM variant.

Table 4: BLEU scores on the Multi30K validation set. For the attention type, Emb denotes word embeddings, $x$H denotes a $x$-dimensional slice of the hidden state (300H corresponds to the whole hidden state), and Emb+$x$H denotes the concatenation of the two.

| Model | Attention | 1 bit | 2 bit | 3 bit | 5 bit | No Limit |
|-------|-----------|-------|-------|-------|-------|----------|
| RevLSTM | 20H | 28.51 | 29.72 | 30.65 | 29.82 | 29.11 |
| | 100H | 28.10 | 35.52 | 36.13 | 34.97 | 35.14 |
| | 300H | 26.46 | 36.73 | 37.04 | 37.32 | 37.27 |
| | Emb | 31.27 | 30.96 | 31.41 | 31.31 | 31.95 |
| | Emb+20H | 36.33 | 36.75 | 37.54 | 36.89 | 36.51 |
| RevGRU | 20H | 25.96 | 25.86 | 27.25 | 27.13 | 26.96 |
| | 100H | 32.52 | 32.86 | 31.08 | 31.16 | 31.87 |
| | 300H | 34.26 | 34.00 | 33.02 | 33.08 | 32.24 |
| | Emb | 27.57 | 27.59 | 28.03 | 27.24 | 28.07 |
| | Emb+20H | 33.67 | 34.94 | 34.36 | 34.87 | 35.12 |

**IWSLT-2016 Experiments.** For both the encoder and decoder we used unidirectional, two-layer RNNs with 600-dimensional hidden states and 600-dimensional word embeddings. We applied variational dropout with probability 0.4 to the inputs and the output of each layer. The learning rates were initialized to 0.2 for the GRU, RevGRU, and RevLSTM, and 1 for the LSTM. We used the same learning rate decay and stopping criterion as for the Multi30K experiments.

The RevGRU with attention over the concatenation of embeddings and a 60-dimensional slice of the hidden state and 5 bit forgetting achieved a BLEU score of 23.65 on the IWSLT validation set; the RevLSTM with the same attention and forgetting configuration achieved a validation BLEU score of 26.17. The baseline GRU achieved a validation BLEU score of 18.92, while the baseline LSTM achieved 26.31.

# H  Memory Savings

## H.1  Language modeling

### 1 layer RevGRU on Penn TreeBank

Ratio of memory used by storing discarded information in a buffer and using reversibility vs. storing all activations naïvely. **Left:** Actual savings obtained by our method. **Right:** Idealized savings obtained by using a perfect buffer.

**2 layer RevGRU on Penn TreeBank**

Ratio of memory used by storing discarded information in a buffer and using reversibility vs. storing all activations naïvely. **Left:** Actual savings obtained by our method. **Right:** Idealized savings obtained by using a perfect buffer.

**1 layer RevLSTM on Penn TreeBank**

Ratio of memory used by storing discarded information in a buffer and using reversibility vs. storing all activations naïvely. **Left:** Actual savings obtained by our method. **Right:** Idealized savings obtained by using a perfect buffer.

**2 layer RevLSTM on Penn TreeBank**

Ratio of memory used by storing discarded information in a buffer and using reversibility vs. storing all activations naïvely. **Left:** Actual savings obtained by our method. **Right:** Idealized savings obtained by using a perfect buffer.

## H.2 Neural Machine Translation

In this section, we show the memory savings achieved by the encoder and decoder of our reversible NMT models. The memory savings refer to the ratio of the amount of memory needed to store discarded information in a buffer for reversibility, compared to storing all activations. Table 5 shows the memory savings in the decoder for various RevGRU and RevLSTM models on Multi30K.

Table 5: Average memory savings in the decoder for NMT on the Multi30K dataset, during training. For the attention type, Emb denotes word embeddings, $x$H denotes a $x$-dimensional slice of the hidden state, and Emb+$x$H denotes the concatenation of the two.

| Model | Attention | 1 bit | 2 bit | 3 bit | 5 bit | No Limit |
|---|---|---|---|---|---|---|
| RevLSTM | 20H | 24.0 | 13.6 | 10.7 | 7.9 | 6.6 |
| | 100H | 24.1 | 13.9 | 10.1 | 8.0 | 5.5 |
| | 300H | 24.7 | 13.4 | 10.7 | 8.3 | 6.5 |
| | Emb | 24.1 | 13.5 | 10.5 | 8.0 | 6.7 |
| | Emb+20H | 24.4 | 13.7 | 11.1 | 7.8 | 7.8 |
| RevGRU | 20H | 24.1 | 13.5 | 11.1 | 8.8 | 7.9 |
| | 100H | 26.0 | 14.1 | 12.2 | 9.5 | 8.2 |
| | 300H | 26.1 | 14.8 | 13.0 | 10.0 | 9.8 |
| | Emb | 25.9 | 14.1 | 12.5 | 9.8 | 8.3 |
| | Emb+20H | 25.5 | 14.8 | 12.9 | 11.2 | 8.9 |

In sections H.2.1, H.2.2, and H.2.3, we plot the memory savings during training for RevGRU and RevLSTM models on Multi30K and IWSLT-2016, using various levels of forgetting. In each plot, we show the actual memory savings achieved by our method, as well as the idealized savings obtained by using a perfect buffer.

### H.2.1 RevGRU on Multi30K

Figure 1: **RevGRU 20H.** From left to right: 1 bit, 3 bits, and no limit on forgetting.

Figure 2: **RevGRU 100H.** From left to right, 1 bit, 3 bits, and no limit on forgetting.

Figure 3: **RevGRU Emb+20H.** From left to right: 1 bit, 3 bits, and no limit on forgetting.

### H.2.2 RevLSTM on Multi30K

Figure 4: **RevLSTM 20H.** From left to right: 1 bit, 3 bits, and no limit on forgetting.

Figure 5: **RevLSTM 100H.** From left to right: 1 bit, 3 bits, and no limit on forgetting.

Figure 6: **RevLSTM Emb+20H.** From left to right: 1 bit, 3 bits, and no limit on forgetting.

### H.2.3 RevGRU and RevLSTM on IWSLT-2016

Here, we plot the memory savings achieved by our two-layer models on IWSLT-2016, as well as the ideal memory savings, for both the encoder and decoder.

Figure 7: Memory savings on IWSLT. **Left:** RevGRU. **Right:** RevLSTM. Both models use attention over the concatenation of the word embeddings and a 60-dimensional slice of the hidden state.

# I  Training/Validation Curves

## I.1  Penn TreeBank

### 1 layer RevGRU

Training/validation perplexity for a 1-layer RevGRU on Penn TreeBank with various restrictions on forgetting and a baseline GRU model. **Left:** Perplexity on the training set. **Right:** Perplexity on the validation set.

### 2 layer RevGRU

Training/validation perplexity for a 2-layer RevGRU on Penn TreeBank with various restrictions on forgetting and a baseline GRU model. **Left:** Perplexity on the training set. **Right:** Perplexity on the validation set.

## 1 layer RevLSTM

Training/validation perplexity for a 1-layer RevLSTM on Penn TreeBank with various restrictions on forgetting and a baseline LSTM model. **Left:** Perplexity on the training set. **Right:** Perplexity on the validation set.

## 2 layer RevLSTM

Training/validation perplexity for a 1-layer RevLSTM on Penn TreeBank with various restrictions on forgetting and a baseline LSTM model. **Left:** Perplexity on the training set. **Right:** Perplexity on the validation set.

## I.2 WikiText-2

### 1 layer RevGRU

Training/validation perplexity for a 1-layer RevGRU on WikiText-2 with various restrictions on forgetting and a baseline GRU model. **Left:** Perplexity on the training set. **Right:** Perplexity on the validation set.

### 2 layer RevGRU

Training/validation perplexity for a 2-layer RevGRU on WikiText-2 with various restrictions on forgetting and a baseline GRU model. **Left:** Perplexity on the training set. **Right:** Perplexity on the validation set.

**1 layer RevLSTM**

Training/validation perplexity for a 1-layer RevLSTM on WikiText-2 with various restrictions on forgetting and a baseline LSTM model. **Left:** Perplexity on the training set. **Right:** Perplexity on the validation set.

**2 layer RevLSTM**

Training/validation perplexity for a 2-layer RevLSTM on WikiText-2 with various restrictions on forgetting and a baseline LSTM model. **Left:** Perplexity on the training set. **Right:** Perplexity on the validation set.

## I.3 Multi30K NMT

In this section we show the training and validation curves for the RevLSTM and RevGRU NMT models with various types of attention (20H, 100H, 300H, Emb, and Emb+20H) and restrictions on forgetting (1, 2, 3, and 5 bits, and no limit on forgetting). For Multi30K, both the encoder and decoder are single-layer, unidirectional RNNs with 300 hidden units.

### I.3.1 RevGRU

Figure 8: **RevGRU 20H** (attention over a 20-dimensional slice of the hidden state).

Figure 9: **RevGRU 100H** (attention over a 100-dimensional slice of the hidden state).

Figure 10: **RevGRU 300H** (attention over the whole hidden state).

Figure 11: **RevGRU Emb** (attention over the input word embeddings).

Figure 12: **RevGRU Emb+20H** (attention over a concatenation of the word embeddings and a 20-dimensional slice of the hidden state).

### I.3.2    RevLSTM

Figure 13: **RevLSTM 20H** (attention over a 20-dimensional slice of the hidden state).

Figure 14: **RevLSTM 100H** (attention over a 100-dimensional slice of the hidden state).

Figure 15: **RevLSTM 300H** (attention over the whole hidden state).

Figure 16: **RevLSTM Emb** (attention over the input word embeddings).

Figure 17: **RevLSTM Emb+20H** (attention over a concatenation of the word embeddings and a 20-dimensional slice of the hidden state).

## I.4 IWSLT 2016

Figure 18: Training/validation perplexity for a 2-layer, 600-hidden unit encoder-decoder architecture, with attention over a 60-dimensional slice of the hidden state, and 5 bit forgetting. **Left:** RevGRU. **Right:** RevLSTM.

## Footnotes

[1]When $h_i$ is negative, we must perform an additional step of appending ones to the bit representation of $h_i$ due to using two's complement representation.

[2]Note that the Softmax is computed over rows, so the first dimension of the matrix $Q$ must be $FH$.