[Reviews · NeurIPS 2018]

Reviewer 1



Summary: Authors investigate reversible architectures for RNN to reduce the memory requirement of their training. They build upon the reversible residual network architecture. By allowing reversibility, RNN can recompute the hidden activation during the backpropagation through time, trading computation for memory. Authors argue that a naïve reversible RNNs are not able to forget, hence have trouble to solve some simple learning tasks. To tackle this issue, they propose a variant, that store the lost information due to the forgetting. Authors then perform an extensive empirical evaluation on Penn TreeBank, Wikitext-2, Multi30K and IWLST, where they show that the reversible GRU/LSTM are on part and sometime outperform their non-reversible counterparts. Clarity, The paper is clearly written and pleasant to read overall. Section 4 clarity could be improved. It would be informative to report more details about the experiments (model architectures (size of the hidden states), dataset splits..) and what are the final performances of the different model. In addition, the statement ‘the remember task for an ordinary model with forgetting reduces to the repeat task for a perfectly reversible RNN’ is confusing to me? Quality, Authors perform an extended empirical evaluation to validate their claims on several language modeling and machine translation tasks. My main concern is that the authors do not report previous works scores on those different tasks. Therefore, it is not clear if the baselines studied in this paper are competitive with respect to previous work. Few remarks: - While section 4 results indicates the memorization capability of NF-RevGRU/NF-RevLSTM, it does not really show their ‘Impossibility of Zero Forgetting’ ? - Given a fix hidden state size, a reversible architecture have more parameters than its non-reversible counterpart. Is this factor taken into account in the comparison? - It would be informative to reports that additional computational cost in addition to the memory saving. Originality: To my knowledge, this is the first work exploring reversible architecture for recurrent neural networks. Significance: The memory cost associated with training RNN is an important limitation. Tackling this issue could allow the exploration of larger RNN models. Other approaches have try to reduce the memory cost associated with training a RNN, by investigating real-time recurrent learning or recurrent backpropagation (see ‘Unbiased Online Recurrent Optimization (Tallec et al., 2017) or ‘Reviving and Improving Recurrent Back-Propagation’ (Liao, 2018)). How does reversible architecture compare in term of computation/memory cost to those approaches? Overall, I think the paper tackles an important problem and I appreciate the extended empirical evaluation. For those reason I lean toward acceptance. Some aspect of the paper could be improved as reporting previous work performances on the different tasks and a comparison with other approach that also tackle the problem of RNN training memory cost. After Rebuttal: Thanks for the feedback, as it addresses some of my concern, I will revise my score accordingly. Also it would be nice to refer to state-of-art model for the different task in the main text for completeness.

Reviewer 2



This paper introduces reversible recurrent neural network. The main idea behind reversible networks is their ability to reconstruct the hidden state at time (or layer) t using the hidden state (or layer) t+1. This allows to recompute all the hidden state during back-propagation, without having to store anything, thus allowing big memory savings for training RNNs. The main contribution of this paper is to propose variants of LSTM and GRU which have this property. As discussed by the authors (and empirically observed in previous work), being able to "forget" is important for RNNs on tasks such as language modeling. However, the reversibility property also means that the RNNs do not forget (if you can reconstruct all hidden state from the last, you can predict the whole sequence). The solution considered in the paper is to allow the RNNs to forget a limited amount of information (2-5 bits per hidden units), and store this information for back-propagation. The method is evaluated on two language modeling benchmarks as well as machine translation. This paper is very clear and well written, and I enjoyed reading it. In particular, the discussion regarding the impossibility of zero forgetting, as well as the link between finite precision, forget gate values and forgetting was interesting. The experimental evaluation of the method is thorough, and done on multiple benchmarks (in particular, showing how this can be applied with attention models is a big plus). My main nitpicking regarding the experimental results is that the authors did not include regular LSTM/GRU as baseline in the papers, and the reported results seem quite a bit worse than results for similar models reported in previous work. I am also wondering what is the impact of using dropout with reversible neural networks, and believe that the authors should add a discussion about this. Overall, while the paper builds on existing work, and the method is similar to Gomez et al. (2017), I believe the contributions to be significant enough for acceptance to NIPS. Indeed, the forgetting issue which is crucial for RNNs (and is the focus of this work), was not discussed as far as I know in previous papers (Gomez et al. applied this method to CNN for computer vision). To conclude, the paper is clearly written, technically sound and the contributions are different enough from previous work for acceptance to NIPS. I will increase my score based on the answers to my questions (baseline and dropout) after the rebuttal. UPDATE: After the rebuttal, I have decided to update my score to 7. I suggest to put the performance of the baseline models in the tables, as it make it easier to compare them to the ones of the proposed method.

Reviewer 3



SUMMARY The paper extends recent works on reversible networks (e.g. RevNets) to RNNs/GRUs/LSTMs. The paper is more interesting than an incremental extension because it explores a fundamental tradeoff between the model's ability to forget using the gating mechanism and the reversibility of its computation. The main contributions of the paper are: 1. variants of GRUs and LSTMs that allow for reversible computing (similar to RevNets), 2. an explicit discussion of the tradeoff mentioned above, 3. an adaptation of the reversible multiplication of Maclaurin et al. for this problem, and 4. empirically matching the performance of standard GRUs/LSTMs with the proposed reversible counterparts while using 5-15 times less memory. STRENGTHS - The paper is very clear and well-written. It does a great job of motivating the problem, explaining why it is nontrivial, and presents a sensible solution that works. - The discussion on the forgetting-reversibility tradeoff, albeit hampered by the confusing example (see below), is insightful and may inspire additional research directions. - The scheme for storing lost information is effective in practice. For instance, the validation perplexity in PTB LM by a standard LSTM is 73 (Table 3) whereas the one by a reversible version is 72.9 (Table 1) with close to 5 times less memory usage. WEAKNESSES - I wasn't fully clear about the repeat/remember example in Section 4. I understand that the unrolled reverse computation of a TBPTT of an exactly reversible model for the repeat task is equivalent to the forward pass of a regular model for the remember task, but aren't they still quite different in major ways? First, are they really equivalent in terms of their gradient updates? In the end, they draw two different computation graphs? Second, at *test time*, the former is not auto-regressive (i.e., it uses the given input sequence) whereas the latter is. Maybe I'm missing something simple, but a more careful explanation of the example would be helpful. Also a minor issue: why are an NF-RevGRU and an LSTM compared in Appendix A? Shouldn't an NF-RevLSTM be used for a fairer comparison? - I'm not familiar with the algorithm of Maclaurin et al., so it's difficult to get much out of the description of Algorithm 1 other than its mechanics. A review/justification of the algorithm may make the paper more self-contained. - As the paper acknowledges, the reversible version has a much higher computational cost during training (2-3 times slower). Given how cheap memory is, it remains to be seen how actually practical this approach is. OTHER COMMENTS - It'd still be useful to include the perplexity/BLEU scores of a NF-Rev{GRU, LSTM} just to verify that the gating mechanism is indeed necessary. - More details on using attention would be useful, perhaps as an extra appendix.